# Evaluating liver type fatty acid binding protein as a diagnostic and prognostic biomarker in metabolic dysfunction-associated steatotic liver disease in pediatric patients

Nadia Al Mazrouei[1], Mostafa Mohsen ElGharbawy[2], Mahitab Gamal[3], Shaimaa Emad[4],
Amal Ahmed Mohamed[5], Osama Mohamed Ibrahim[6], Asim Ahmed Elnour[7],
Marafi Jammaa Ahmed [8]*, Semira Abdi Beshir[9], Vineetha Menon[10],
Sami Fatehi Abdalla[11,12], Ali Awadallah Saeed[13], Heba Ali ElRamly[14]

**1** Pharmacy Practice and Pharmacotherapeutics department, University of Sharjah, United Arab Emirates,
**2** Clinical Pharmacy department, New Giza University, Egypt, **3** Pharmacology department, New Giza
University, Egypt, **4** Fellow of pediatric Hepatology, National Hepatology and Tropical Medicine Research
Institute, Cairo, Egypt, **5** Biochemistry and Molecular Biology Department, National Hepatology and
Tropical Medicine Research Institute, Cairo, Egypt, **6** Clinical Pharmacy department, New Giza University,
**7** Program of Clinical Pharmacy, College of Pharmacy, Al Ain University, Abu Dhabi campus, Abu
Dhabi-United Arab Emirates (UAE). AAU Health and Biomedical Research Center, Al Ain University, Abu
Dhabi, United Arab Emirates, **8** Faculty of Medicine, Bahri University, Khartoum, Sudan, **9** Department
of clinical pharmacy and Pharmacotherapeutics, Dubai Pharmacy College for Girls, Dubai, United Arab
Emirates (UAE), **10** Assistant Professor, Department of Pharmacy Practice, College of Pharmacy,
Gulf Medical University, United Arab Emirates (UAE), **11** Department of Clinical Sciences, College of
Medicine, Almaarefa University, Daryiyah, Saudi Arabia, **12** Research Center, Deanship of Scientific
Research and Post-Graduate Studies, AlMaarefa University, Diriyah, Riyadh, Saudi Arabia, **13** Department
of Pharmacology, Faculty of Clinical and Industrial Pharmacy, National University-Sudan, Mycetoma
Research Center, Khartoum, Sudan, **14** Clinical Pharmacy Department, New Giza University, Egypt

* marafi.ahmed99@gmail.com

## Abstract

### Background

Metabolic dysfunction-associated steatotic liver disease (MASLD), is a common liver
disorder, predicted to increase globally. Currently, non-invasive methods are pro-
posed for the assessment of (MASLD). This study aimed to investigate the use of
serum Liver Type Fatty Acid Binding Protein (L-FABP) concentration as a diagnostic
and prognostic biomarker in pediatric MASLD patients and to evaluate its relationship
with steatosis and fibrosis.

### Methods

An observational, cross-sectional study on paediatric MASLD patients. Serum levels
of L-FABP and hepatic biochemical markers were measured and analyzed. Statistical
analyses evaluated the association between L-FABP and other markers, while logis-
tic regression and ROC curves assessed its diagnostic accuracy.

org/10.1371/journal.pone.0333581

della Campania Luigi Vanvitelli Scuola di
Medicina e Chirurgia, ITALY

**Peer Review History:** PLOS recognizes the
benefits of transparency in the peer review
process; therefore, we enable the publication
of all of the content of peer review and
author responses alongside final, published
articles. The editorial history of this article is
available here: https://doi.org/10.1371/journal.
pone.0333581

**Data availability statement:** All relevant data are within the paper and its Supporting Information files.

**Funding:** The author(s) received no specific funding for this work.

**Competing interests:** The authors have declared that no competing interests exist.

## Results

Serum L-FABP levels showed a significant difference between the MASLD and control groups *(P<0.0001).* The results of logistic regression revealed that each one-unit elevation of L-FABP level was associated with 144.5% higher odds of being MASLD (95% CI: 129.3% − 167.8%). A receiver operating characteristics (ROC) curve was constructed to assess the diagnostic accuracy of L-FABP. The resulted area under the ROC curve (AUC) was 0.885. The cutoff value was 5.7 ng/ mL, with sensitivity of 72.73%, and a specificity of 93.62%. The results also showed that the odds ratio of progressing from stage F2 to F3 increases by 108.9% (95% CI: 87.2% − 141.4%) for each one-unit increase in serum L-FABP level.

## Conclusion

L-FABP shows promise as a non-invasive biomarker for diagnosing and monitoring MASLD in pediatric patients. Its association with disease stages suggests its utility in assessing disease progression, particularly in advanced stages.

## I. Introduction

Metabolic dysfunction-associated steatotic liver disease (MASLD), previously termed non-alcoholic fatty liver disease (NAFLD), is defined by the presence of hepatic steatosis in more than 5% of hepatocytes, often linked to insulin resistance. It represents a form of steatotic liver disease (SLD) associated with one or more cardiometabolic risk factors, in the absence of significant alcohol consumption. MASLD encompasses a spectrum ranging from simple steatosis to metabolic dysfunction-associated steatohepatitis (MASH, formerly NASH), progressive fibrosis, cirrhosis, and MASH-related hepatocellular carcinoma (HCC). [1–3] The prevalence of Metabolic dysfunction-associated steatotic liver disease (MASLD) varies globally, ranging from 25 to 29%. It is predicted to reach 30.7% by 2040 [4,5]. North Africa and Middle East have one of the highest prevalence rates of MASLD in general, with an estimation of 31.8% of all adults being influenced. One of the top 10 countries for obesity prevalence is Egypt yet there are no exact stats for the MASLD NAFLD prevalence [6,7]. MASLD prevalence in children with obesity is about 52.49%, and given Egypt's high childhood obesity rates, the number of affected children is expected to be significant. [2–4]. There is limited research on the epidemiology of MASLD in Egypt. However, a study conducted in Alexandria reported a 15.8% prevalence of MASLD among schoolchildren, with a significant increase as they aged. Additionally, the prevalence of MASLD among healthy Egyptian college students was found to be 31.8% [8].

In adults, MASLD is anticipated to become a leading cause of liver transplantation. However, in pediatrics, the primary concern is the progression of fibrosis and cirrhosis in early adulthood.

This augments the need to have an early noninvasive diagnostic biomarker that might be valuable in detecting progression [9,10].

Although ALT and AST are frequently elevated in MASLD, they alone are not sufficient for diagnosis, as some patients, particularly those with mild steatosis, may have normal levels. More advanced fibrosis markers, such as the FIB-4 index, MASLD fibrosis score, or Enhanced Liver Fibrosis (ELF) test, are increasingly used to assess disease severity. However, the utility of these tests in pediatric MASLD is still under investigation [11,12].

The gold standard for differentiating MASLD from Metabolic Dysfunction-Associated Steatohepatitis (MASH) previously known as Nonalcoholic Steatohepatitis (NASH) remains the liver biopsy. However, this method has significant limitations. It's invasive, expensive, and carries the risk of sampling errors, serious complications and pain. Furthermore, there is substantial intra- and interobserver variability, even among expert pathologists. Additionally, patients, especially children, are also reluctant to undergo this procedure. Therefore, there is a growing need for noninvasive methods to diagnose the different stages of MASLD [13–17].

Liver-type fatty acid-binding protein (L-FABP) is an antioxidant protein endogenously synthesized in the liver and the proximal tubular epithelial cells of the kidney, but not in skeletal or cardiac muscle. Hepatocytes produce L-FABP, as an essential regulator for metabolism of fats in the normal liver. Advanced liver disease is highly correlated with elevated L-FABP expression in liver which is directly reflected on the serum L-FABP [18–20]. Hepatocellular damage is a key driver of chronic liver disease progression, linked to various causes, including NAFLD and NASH, ASH, hepatitis C, and hepatitis B. Serum L-FABP levels are used to monitor liver damage in NAFLD, fibrosis in NASH, and HCV infection [20]. A study measuring serum L-FABP in NASH patients and healthy controls found significant correlations with the NAFLD activity score (NAS), fibrosis, inflammation, and serum ALT levels. These findings suggest serum L-FABP is a valuable biomarker from early NAFL to advanced NASH stages [20].

Recent studies have highlighted L-FABP as a promising biomarker due to its sensitivity and reliability. It is considered as a prognostic marker for predicting the survival probability in both acute and chronic liver diseases as well as assessing the advancement of liver diseases in diverse stages as MASLD can progress through stages of fibrosis, with some patients developing cirrhosis, with potential complications, including HCC [20–22].

This study aimed to investigate the association between serum L-FABP levels and pediatric MASLD, and to evaluate its diagnostic and prognostic utility, with particular emphasis on fibrosis severity.

This study is pioneering in its evaluation of L-FABP concentration as a prognostic and diagnostic marker in the pediatric population.

## II. Patients and methods

### II. 1 Ethical consideration

The study was carried out according to the Good Clinical Practice (GCP) guidelines and the Declaration of Helsinki. The study protocol was approved by the Institutional Review Board of the National Hepatology and Tropical Medicine Research Institute (NHTMRI), approval serial (31−2022), and the Ethics Committee Board of New Giza University (approval number CP-009). The child's legally acceptable guardian signed the informed written consent form.

### II. 2 Patients and study design

This was an observational cross-sectional study that included 100 Egyptian MASLD pediatric patients and 99 healthy children as a control group.

Sample size was calculated using GPower® software. Considering logistic regression model with L-FABP as a single normally distributed continuous predictor (μ = 6.5, σ = 2), and a binary response (MASLD or not), a total sample size of 199 was required to test an effect size represented by odds ratio = 1.5 in a two-sided test with α = 0.05 at a power of at least 85%. The total number was divided between MASLD and control groups at equal ratio.

The study participants comprised Egyptian paediatric patients with MASLD who were treated at the National Hepatology and Tropical Medicine Research Institute. The recruitment period was from 20/10/2022 to 30/06/2023. Participants were

selected through convenience sampling from clinic visitors and their accompanying relatives, with eligibility determined by their willingness to provide informed consent and biological samples. The control group consisted of healthy children with no clinical signs of MASLD. Controls underwent appropriate assessments, including ultrasound imaging, to exclude hepatic steatosis, given its asymptomatic nature. All patients received standard care and treatment as prescribed by their physicians, without intervention from the study team. None of the participants were on ursodeoxycholic acid (UDCA), vitamin E, or other hepatoprotective medications at the time of enrollment. As pharmacologic therapies are not currently approved for pediatric MASLD, management primarily involved lifestyle interventions such as dietary modifications, physical activity, and gradual weight loss where appropriate. The **inclusion criteria** were: Asymptomatic patients or with a history of Metabolic dysfunction-associated steatotic liver disease. Normal international normalised ratio (INR). Age was > 2.5 years to <18 years, Confirmed MASLD cases with predefined staging. On the other hand, children with fulminant or acute liver failure (AFL), (increased INR > 1.5), cirrhosis, elevated transaminases (indicating acute liver disease), confirmed HCV, HBV and HCC, children with suspected or confirmed genetic/metabolic liver diseases (e.g., Wilson's disease, alpha-1 antitrypsin deficiency, glycogen storage diseases) or Children whose guardians refused to give an informed consent were excluded.

## II. 3. Blood sample collection and processing

Blood samples were collected under fasting conditions from the participants using yellow gel tubes which contain a clot activator and a gel separator. After centrifugation at 20 °C for 10 minutes at speed of 1500 RPM, the blood cells were separated from the serum. This method prevents contamination and preserves the integrity of the serum for accurate testing results.

Alanine transaminase (ALT), aspartate aminotransferase (AST), gamma-glutamyl transferase (GGT), triglycerides (TG), high-density lipoproteins (HDL-C), total cholesterol (TC), Albumin (ALB), T.Bilirubin (T.Bil) and Glycated haemoglobin (HbA1C) were measured using Olympus AU 400 (Automated biochemistry analyser). Low Density Lipoprotein (LDL-C) were calculated by Fried Ewald Equation for LDL-C. Glycated haemoglobin (HbA1C) was detected using turbidimetric inhibition immunoassay (TINIA) that is validated by the National Glycohemoglobin Standardization Program (NGSP). Platelet count (PLTs) was determined using Beckman Coulter DXH900 Haematology Analyzer. Alpha fetoprotein (AFP) was determined using Eliza kits for all participants (CanAg AFP EIA 600−10). L-FABP was analysed using ELISA Assay (Cat. No E2159Hu, Shanghai, China) according to manufacture instructions.

## II. 4. Abdominal examination

All the participants had an abdominal ultrasound examination using Siemens G50, 5-MHz transducer, Siemens, Munich, Germany. The liver size, contour, echogenicity, posterior beam attenuation in addition to the structure were all evaluated. The extent of fibrosis and the level of steatosis was based on liver brightness in addition to a transient elastography fibroscan, using the FibroScan® Compact 530 (France) [23–25]. Fibro Scan measures liver stiffness in kilopascals (kPa), correlating with fibrosis severity. These classify fibrosis on a scale of **F0 to F3**.
**F0**: - **F1**: Mild perisinusoidal or periportal fibrosis (< 5.0–7.0 kPa). **F2**: Moderate fibrosis with portal/periportal involvement (7.1–9.6 kPa). **F3**: Bridging fibrosis (progressive scarring but no cirrhosis) 9.7–12.5 kPa

During the abdominal ultrasound examination, specific hepatic and splenic features were systematically assessed. The following sonographic findings were evaluated and documented:

- **Bright hepatomegaly**: Refers to an enlarged liver with increased echogenicity, suggesting hepatic steatosis or fibrosis.

- **Coarse liver**: Indicates heterogeneous and irregular parenchymal echotexture, often associated with chronic liver disease or advanced fibrosis.

- **Fatty liver**: Characterized by diffuse hyperechogenicity of the liver parenchyma, with posterior beam attenuation, consistent with hepatic steatosis.

- **Splenomegaly**: Enlargement of the spleen, which may reflect portal hypertension or chronic liver pathology.

## II. 5. Statistical analysis

SAS® software (Academic version) was used for statistical analyses. Measurements were presented as mean ± standard deviation (mean ±SD) for numeric variables and number (%) for categoric variables. Regression analysis (using the GLM procedure) was conducted to evaluate the relationship between the L-FABP level and hepatic biochemical markers. Binary logistic regression was applied to assess the use of serum L-FABP level as a predictor of MASLD and fibrosis stages in the study population and receiver operating characteristic curve (ROC) was constructed based on the logistic model to assess the diagnostic accuracy of L-FABP. The best L-FABP concentration cut-off value for diagnosing MASLD was determined using Youden index [26]. A P-value of < 0.05 was regarded as statistically significant for all tests.

## III. Results

### III. 1. Patients' demographics and laboratory data

The results of the laboratory tests of patients in each study group are presented in Table 1. Among the total participants in the study, there were 101 males (50.75%) and 98 females (49.25%). Analysis of the data in Table 1 indicates a significant difference between the control and MASLD groups in terms of liver fatty acid-binding protein (L-FABP), alanine amino-transferase (ALT), aspartate aminotransferase (AST), albumin, total cholesterol (TC), triglycerides (TG), body mass index (BMI), gamma-glutamyl transferase (GGT), alpha-fetoprotein (AFP), hemoglobin A1C (HbA1C), high-density lipoprotein (HDL), low-density lipoprotein (LDL), fasting blood sugar (FBS), total bilirubin (T.BIL). However, age, platelets (PLTS), serum creatinine and hemoglobin did not exhibit a significant difference between the two groups.

### III. 2. Ultrasonographic examination and fibrosis stages

Ultrasound examination was performed as a routine initial imaging modality to assess hepatic echogenicity, size, and texture [27–29]. The results revealed that 27 patients (27%) showed bright hepatomegaly, 38 patients (38%) showed coarse liver, 31 patients (31%) showed fatty liver, and 4 patients (4%) showed splenomegaly, (Fig 1). "Bright hepatomegaly" indicates increased echogenicity and hepatomegaly, whereas "fatty liver" refers to uniform hepatic steatosis with preserved size. "Coarse liver" denotes advanced fibrosis but does not imply cirrhosis. Regarding the fibrosis stage, 66 patients (66%) were in F0-F1 stage, 20 patients (20%) were in F2 stage, and 14 patients (14%) were in F3 stage.

### III. 3. Association of Serum L-FABP With Liver biochemical markers and Other Markers

The results in Table 2 represent the assessment of possible association between the L-FABP level and patients' lab results. It covers a range of variables such as age, liver biochemical markers (ALT, AST, Albumin, GGT, T.Bil), metabolic indicators (BMI, HbA1C, FBS, TG, TC, LDL, HDL), and markers of liver health or disease (AFP, PLTs, Fibrosis Stages, Sonography). This was assessed by univariable and multivariable regression analysis. Only terms that showed a significant association with L-FABP in the univariable analysis (taking the group into consideration) were subjected to multivariable analysis to adjust for the presence of other variables. In univariable analysis, ALT, Albumin, T. BIL, PLTs, age and HbA1C showed significant association with the L-FABP (*P-value* <0.05). However, after adjusting for other patient values in multivariable analysis, only ALT and HbA1C showed significant association (*P-value* <0.05) with the L-FABP. Albumin showed a significant association with L-FABP. Serum AST, GGT, FBS, lipid profile, BMI and AFP did not show any significant association with the serum L-FABP.

### III. 4. Association Between L- FABP and Non-Invasive Diagnostic Measures

The association between serum L-FABP and the non-invasive diagnostic methods, Fibrosis stages and Ultrasono-graphic diagnosis, was assessed in the MASLD group since the control group is healthy. Fibrosis stages showed significant association with the serum L-FABP (*P-value <0.0001*) in both univariable and multivariable analysis, while the

**Table 1. Patients' demographic and lab data.**

| Variable | MASLD N = 100 Mean ± SD | Control N = 99 Mean ± SD | P-value |
|---|---|---|---|
| Age (Years) | 12.27 ± 3.05 | 12.11 ± 3.04 | 0.7839 |
| BMI (Kg/m²) | 33.69 ± 4.57 | 23.2 ± 2.24 | <0.0001 |
| L-FABP (ng/ml) | 8.02 ± 3.70 | 3.88 ± 1.41 | <0.0001 |
| ALT (U/L) | 40.96 ± 10.07 | 29.89 ± 5.26 | <0.0001 |
| AST (U/L) | 40.92 ± 6.95 | 32.21 ± 6.26 | <0.0001 |
| Albumin (g/dl) | 3.63 ± 0.37 | 3.86 ± 0.21 | <0.0005 |
| TC (mg/dl) | 187.87 ± 17.21 | 160.61 ± 18.14 | <0.0001 |
| GGT (U/L) | 48.50 ± 16.93 | 33.25 ± 7.35 | <0.0001 |
| AFP (ng/dl) | 11.10 ± 5.24 | 5.87 ± 1.72 | <0.0001 |
| HbA1C (%) | 5.04 ± 0.79 | 3.36 ± 0.85 | <0.0001 |
| HDL (mg/dl) | 36.34 ± 8.94 | 47.29 ± 3.54 | <0.0001 |
| LDL (mg/dl) | 127.23 ± 28.89 | 98.14 ± 6.75 | <0.0001 |
| PLTs (x10³/ µL) | 221.36 ± 78.59 | 233.70 ± 70.14 | 0.40 |
| FBS (mg/dl) | 112.61 ± 17.42 | 92.04 ± 7.84 | <0.0001 |
| T.Bil (mg/dl) | 1.06 ± 0.34 | 0.77 ± 0.18 | <0.0001 |
| TG (mg/dl) | 176.47 ± 30.91 | 144.51 ± 12.86 | <0.0001 |
| Serum Creatinine | 0.91 ± 0.2 | 0.94 ± 0.17 | 0.9394 |
| Haemoglobin | 12.8362 ± 1.15 | 12.1 ± 1.18 | 0.2826 |

*Results are considered significant if (p-value <0.05).*

**Key Abbreviations**

AFP: Alpha fetoprotein; ALT: Alanine transaminase; AST: Aspartate transaminase; BMI: Body mass Index; FBS: Fasting blood sugar; GGT: Gamma-glutamyl transferase; HbA1C: Glycated hemoglobin; HDL-C: High-density lipoproteins; LDL-C: Low density lipoprotein; L-FABP: Liver type fatty acid binding protein; MASLD: Metabolic dysfunction-associated steatotic liver disease; PLTs: Platelets count; T.Bil: Total Bilirubin; TC: Total cholesterol; TG: Triglycerides.

Ultrasonographic diagnosis showed a significant association in the univariable analysis but after adjusting for the fibrosis stages in the multivariable analysis, the relationship was not significant as shown in Table 2, Figs 1 and 2

### III. 5 L-FABP as a predictor of MASLD

Binary logistic regression results revealed that L-FABP is a significant predictor of MASLD (p < 0.0001). The results showed that the odds ratio of being diagnosed with MASLD increases by 144.5% (95% CI: 129.3% − 167.8%) for each one unit increase in serum L-FABP level.

The performance of L-FABP as a diagnostic test for MASLD was assessed using the ROC curve. The resulted area under the ROC curve (AUC) was 0.885 with 95% CI = (0.807–0.94) as shown in Fig 3. The best L-FABP concentration cut-off value for diagnosing MASLD in our study population was determined using Youden index [26], the value was 5.7 ng/ mL, with sensitivity of 72.73%, and a specificity of 93.62%, Fig 3a.

The performance of L-FABP as a diagnostic test for MASLD was compared with other non-invasive scoring systems using ROC curve analysis. The results revealed that L-FABP is a better diagnostic test for MASLD compared to MFS (AUC: 0.783), Fib3 index (AUC: 0.632), Fib 4 index (AUC: 591). The MFS score showed an area that is slightly lower than the AUC of L-FABP, with slightly lower concentration cut-off value (5.2 ng/mL) with lower sensitivity (58%). The diagnostic performance of other scoring systems was obviously inferior to that of L-FABP. The weakest predictor of MASLD among the studied parameters was AST/ALT ratio with area under the ROC curve = 0.564, Fig 3b, 3c,3d and 3e

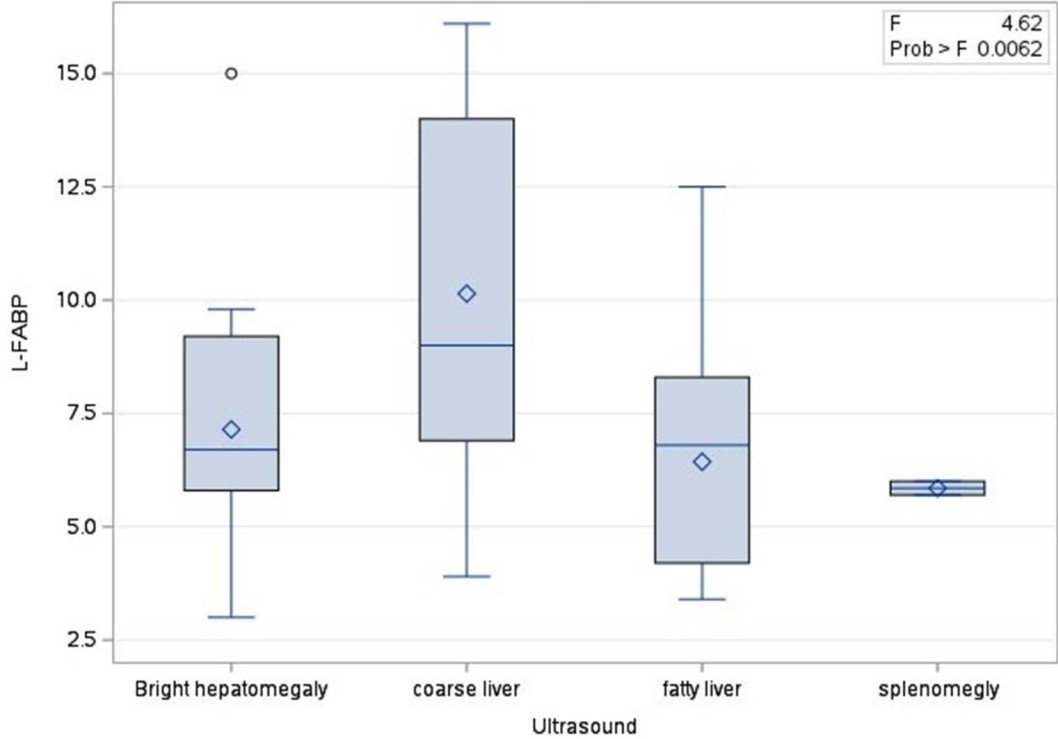

**Fig 1. Distribution of L-FABP in different patients' groups according to ultrasonography.**

### III. 6 L-FABP as a predictor of Fibrosis stages

Binary logistic regression was applied to investigate if the serum L-FABP is a good predictor of the progression of fibrosis Stage from F0-F1 to stage F2. The results revealed that L-FABP was not a significant predictor (p = 0.993), with odds ratio = 0.999 (95% CI: 0.783–1.274).

On the other hand, MASLD was found to be a significant predictor of progression of fibrosis stage from F2 to F3 (p = 0.0183), with odds ratio of being progressed from F2 to F3 increasing by 108.9% (95% CI: 87.2% − 141.4%) for each one unit increase in serum L-FABP level.

## IV. Discussion

Transaminases, especially ALT and AST, play a critical role in monitoring and diagnosing liver diseases. ALT levels rise due to hepatocyte damage, releasing into the bloodstream as a result [15]. Serum concentration of ALT is more closely associated with the onset of MASLD and is more liver-specific than concentrations of AST or Gamma-glutamyl transferase (GGT) [30].

Results in Table 2 demonstrate a significant positive association between ALT and L-FABP across both patient group and control groups, consistent with research indicating a strong association between L-FABP, ALT, and additional liver enzymes [22]. Additionally, research by Chen, Chen, et al., highlighted that, individuals with metabolic syndrome exhibited notably higher ALT levels, which escalated with the accumulation of metabolic syndrome components. This could be due to the fact that elevated ALT levels in MASLD suggest significant liver fat accumulation from various metabolic issues, and they also indicate persistent inflammation that disrupts insulin signaling in the body, including the liver [31–34]. Contrary to expectations, the Dallas Heart Study in the U.S. found that 79% of MASLD patients had normal

**Table 2. Association of patient characteristics with L-FABP serum levels.**

| Variable | Univariable analysis | | Multi-variable analysis P-value |
|---|---|---|---|
| | Pearson's Partial Coefficient | P-value | |
| Age (Years) | 0.287 | 0.004 | 0.5458 |
| BMI (kg/m²) | 0.092 | 0.36 | NA |
| ALT (U/L) | 0.437 | <0.0001 | 0.0001**** |
| AST (U/L) | 0.176 | 0.077 | NA |
| Albumin (g/dl) | −0.543 | <0.0001 | 0.001*** |
| TC (mg/dl) | −0.007 | 0.94 | NA |
| GGT (U/L) | −0.160 | 0.10 | NA |
| AFP (ng/ml) | −0.011 | 0.91 | NA |
| HbA1C (%) | 0.286 | 0.0037 | 0.0001**** |
| HDL (mg/dl) | 0.044 | 0.66 | NA |
| LDL (mg/dl) | −0.048 | 0.63 | NA |
| PLTs (x 10³/µL) | −0.234 | 0.01 | 0.33 |
| FBS (mg/dl) | 0.237 | 0.01 | NA |
| T.Bil (mg/dl) | 0.470 | <0.0001 | 0.313 |
| TG (mg/dl) | −0.038 | 0.70 | NA |
| Fibrosis Stages | NA | <0.0001 | <0.0001**** |
| Ultrasonographic Category | NA | 0.0062 | 0.6143 |

**Keys**

Values are analyzed using SAS software, results are considered significant if (p-value <0.05) ***P<0.0001, ***p<0.001, **p<0.01

Abbreviations: AFP: Alpha fetoprotein; AL T: Alanine transaminase; AST: Aspartate transaminase; BMI: Body mass index; FBS: Fasting blood sugar; GGT: Gamma-glutamyl transferase; HbA1C: Glycated hemoglobin; HDL-C: High-density lipoproteins; LDL-C: Low density lipoprotein; L-FABP: Liver type fatty acid binding protein; MASLD: Metabolic dysfunction-associated steatotic liver disease; PLTs: Platelets count; T.Bil: Total Bilirubin; TC: Total cholesterol; TG: Triglycerides.

ALT levels [35]. Similarly, the Dionysos Study reported that 55% of Italian adults with fatty liver exhibited normal ALT levels [36]. Additionally, research has shown that normal ALT levels can be present even when MASLD is active [36,37]. Moreover, ALT levels vary significantly across the lifespan, influenced by factors such as age, sex, and pubertal status. In children and adolescents, ALT levels tend to rise during puberty, with boys exhibiting higher levels than girls. However, relying solely on ALT levels for diagnosing MASLD in the pediatric population has limitations, as a notable proportion of children with biopsy-confirmed MASLD may present with normal ALT levels. Therefore, while ALT can serve as an initial screening tool, particularly in children over 10 years of age with elevated BMI, it should be complemented with other diagnostic modalities. These may include imaging studies, such as ultrasound or magnetic resonance elastography, and the assessment of additional biomarkers to enhance diagnostic accuracy and effectively monitor disease progression in pediatric MASLD [38]".

Studies indicate that slight elevations in ALT, or values at the higher end of the normal range, could arise from multiple causes. Firstly, the reduction in hepatocyte number and functionality observed in cirrhosis means that serum ALT levels might not accurately reflect the state of hepatocytes. Secondly, ALT's substantial molecular weight (96 KDa) and prolonged half-life could delay its elevation in cases of minor liver damage. Thirdly, liver enzyme levels could be normalized or influenced by treatments with Ursodeoxycholic acid (UDCA) and vitamin E, which may not accurately represent the MASLD condition or its fibrosis stage. Given these considerations, L-FABP emerges as a more advantageous marker due to its smaller molecular size, shorter half-life, specific secretion by the liver, and unaffected status by UDCA treatment, positioning it as a more precise and dependable marker for MASLD prognosis [20,21,39].

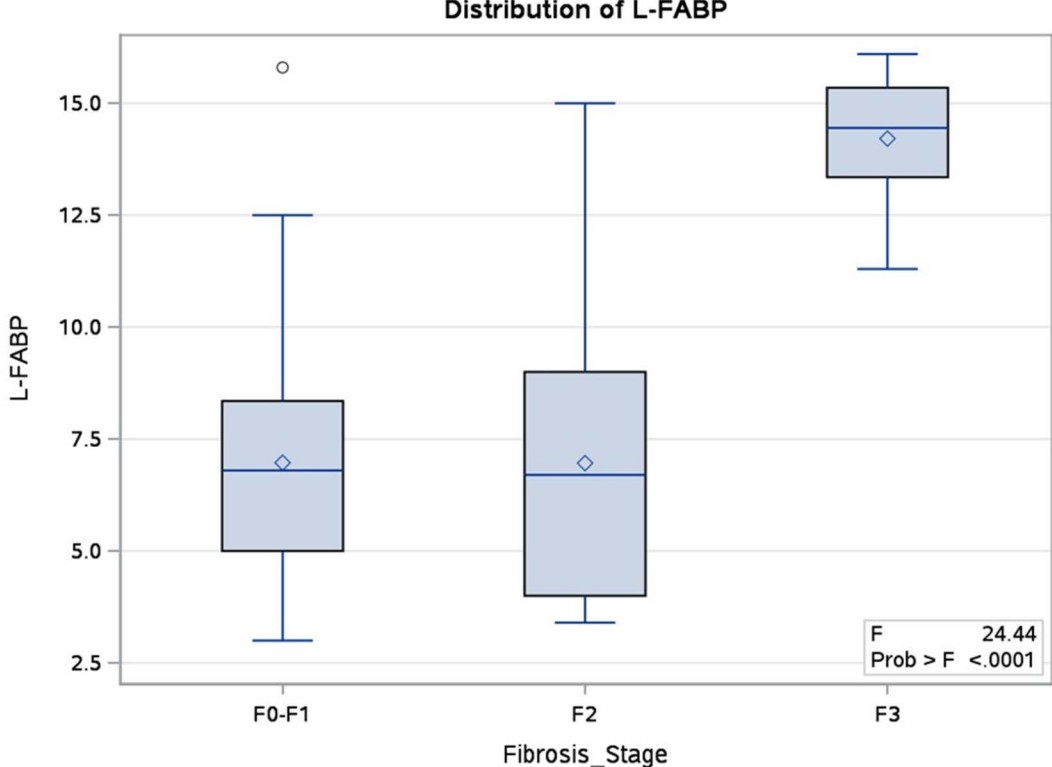

**Fig 2. Distribution of L-FABP in different fibrosis stages.**

Numerous studies have established a robust link between MASLD and metabolic disorders, particularly type 2 diabetes, which both contributes to and exacerbates MASLDprevalence [40,41]. Data from the National Health and Nutritional Examination Survey (NHANES) and other research show MASLD's prevalence at 33.7% generally but rising to 74.9% among diabetics [42–45]. Lu YC et al. (2020) found that elevated FABP1 levels correlate strongly with MASLD in type 2 diabetes patients [46], mirroring the current study findings of significant HbA1C differences between MASLD and control groups, alongside a notable association of HbA1C with L-FABP levels. This underscores the critical need for vigilant monitoring to prevent MASLD progression in non-diabetic or pre-diabetic individuals.

Several studies evaluating MASLD and MASH patient data have noted a significant reduction in serum albumin levels both initially and at the conclusion of the observation period [47,48]. This aligns with our current study's findings, which show association between serum L-FABP and serum albumin in both MASLD and control groups. This may be because albumin specificity increases for MASH, advanced fibrosis, or cirrhosis rather than for MASLD alone. Additionally, as MASLD severity escalates, serum albumin levels tend to decrease [49,50].

Thus, monitoring albumin levels in MASLD patients is advisable for tracking disease progression, as supported by studies showing significant albumin reduction over a two-year period in untreated MASLD cases [50]. Additionally, patients with advanced hepatic fibrosis exhibited notably lower serum albumin compared to those in early stages [51,52].

The present study investigated the association between serum Liver-type Fatty Acid Binding Protein (L-FABP) levels and non-invasive diagnostic and prognostic measures, including fibrosis stages and Ultrasonographic diagnosis, in individuals with MASLD. The results of logistic regression investigating L-FABP as a predictor of MASLD diagnosis and prognosis revealed its potential utility as a biomarker for the diagnosis of MASLD. Moreover, L- FABP was found to be a

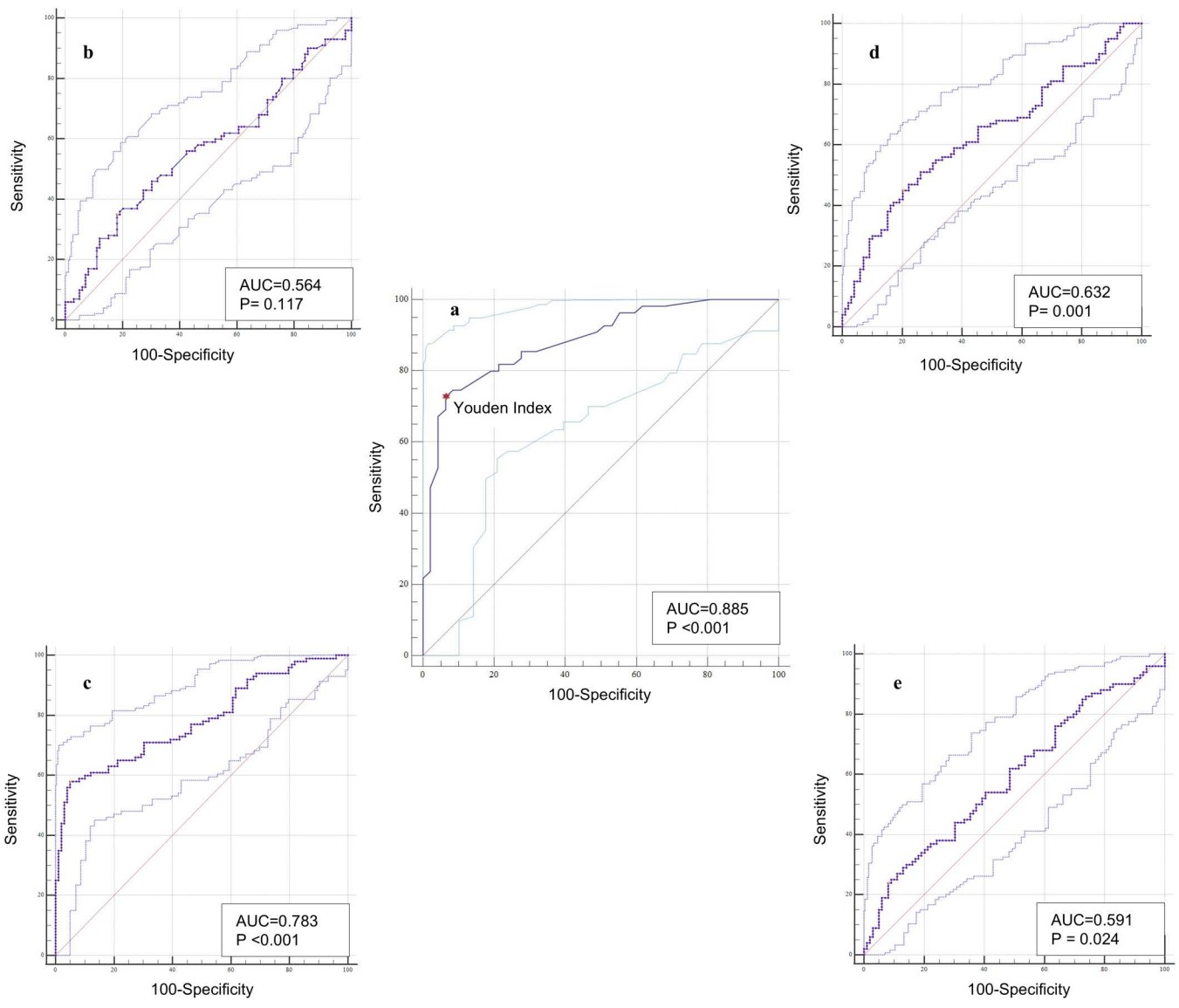

**Fig 3. ROC curve as diagnostic tests for MASLD.** (a) L-FABP, (b) AST/ALT Ratio (c) MFS, (d) Fib3-Index, and (e) Fib-4 index.

good predictor for the disease progression of MASLD from stage F2 to stage F3. However, the results didn't find L-FABP serum level a good predictor for progression from early stage (F0-F1) to stage F2.

The significant association observed between serum L-FABP and fibrosis stages in both univariable and multivariable analyses highlights the clinical relevance of L-FABP in assessing disease progression. This finding aligns with previous study suggesting L-FABP as a promising biomarker for liver injury and metabolic dysfunction associated with MASLD [20,21]. The significant association remained robust even after adjusting for potential confounders, emphasizing the independent predictive value of L-FABP in determining MASLD severity specifically in the advanced stages.

However, the association between serum L-FABP levels and Ultrasonographic diagnosis was only significant in the univariable analysis and lost significance in the multivariable analysis after controlling for fibrosis stages. Ultrasonography, although widely used as a non-invasive diagnostic tool for MASLD, relies on the visual identification

of hepatic steatosis and may not capture the full spectrum of liver injury and metabolic dysfunction represented by L-FABP levels [20].

These findings carry substantial clinical implications. Firstly, serum L-FABP measurement could serve as a valuable adjunctive tool in assessing MASLD severity, providing clinicians with an additional quantitative marker to complement existing diagnostic modalities. Secondly, these results highlight the need for integrated approaches combining biomarkers and imaging techniques such as transient elastography (FibroScan) to achieve better comprehensive evaluation.

The present study's analysis identifies L-FABP as a significant predictor for MASLD, with the binary logistic regression indicating a substantial increase in the odds of MASLD diagnosis associated with serum L-FABP levels. The increase of odds ratio by 144.5% for each unit increase in L-FABP, couples with a highly significant p-value of less than 0.0001, underscores the robustness of L-FABP as a biomarker for MASLD diagnosis.

Furthermore, the diagnostic performance of L-FABP was rigorously assessed using the Receiver Operating Characteristic (ROC) curve, yielding an Area Under the Curve (AUC) of 0.885. This AUC value, with a 95% confidence interval ranging from 0.807 to 0.94, demonstrates good diagnostic accuracy. The optimal cut-off value for L-FABP, determined using the Youden index is 5.7 ng/mL [26]. This threshold provides a balance between sensitivity (72.73%) and specificity (93.62%), suggesting that L-FABP can effectively differentiate between MASLD and non-MASLD individuals within the studied population. In this study, the diagnostic performance of serum L-FABP for detecting MASLD in children was compared to several non-invasive scoring systems, including the MAFLD fibrosis score (MFS), Fib-3 index, Fib-4 index, and the AST/ALT ratio, using ROC curve analysis. L-FABP demonstrated the highest diagnostic accuracy (AUC: 0.885), outperforming MFS (AUC: 0.783), Fib-3 index (AUC: 0.632), Fib-4 index (AUC: 0.591), and AST/ALT ratio (AUC: 0.564). Although MFS showed relatively better performance among the scoring systems, it was still inferior to L-FABP, with lower sensitivity (58%) and a lower L-FABP cut-off (5.2 ng/mL). Unlike composite scores such as Fib-4 or MFS, which rely on indirect markers of liver injury (age, platelet count, ALT/AST), L-FABP is a direct marker of hepatocellular stress and fatty acid metabolism. It is released in response to oxidative stress and inflammation—key drivers in MASLD pathogenesis. Most existing scores are derived and validated in adult populations, with limited reliability in children due to age-related differences in liver enzyme baselines and fibrosis progression. L-FABP's performance is based on actual pediatric data, making it more age-appropriate and clinically relevant. [3,53]

These findings align with previous research, which has highlighted the potential of L-FABP as a diagnostic and prognostic marker in MASLD due to its involvement in lipid metabolism and its elevated expression in various liver conditions [20,54,55]. The strong predictive value of L-FABP for MASLD, as evidenced by the high AUC, reinforces its clinical utility in diagnosing and potentially monitoring the disease progression. L-FABP is a single biomarker, making it easier to implement in clinical practice compared to multi-variable scores requiring multiple lab parameters and complex calculation

## V. Conclusion

MASLD's prevalence in pediatric populations stems from its strong ties to metabolic disorders, encompassing a spectrum of conditions from mild steatosis to severe complications like steatohepatitis and cirrhosis. Unmanaged progression of MASLD can silently escalate to MASH, exacerbating disease. Recent research, including our own findings, highlights a significant association between serum L-FABP levels and key indicators such as ALT, Albumin, and HbA1C, as well as the stages of MASLD. This novel association underscores the potential utility of L-FABP as a biomarker to diagnose and monitor disease progression and fibrosis status, particularly in pediatric cases where early detection is paramount. The study recommends integrating serum L-FABP alongside traditional markers and imaging techniques for diagnosis as well as monitoring MASLD progression in pediatric patients. Moreover, there is a call for further investigation into L-FABP's utility as a novel biomarker across various acute and chronic liver diseases, including MASLD, highlighting its innovative application in pediatric care.

## Strengths and limitations of the study

A major strength is the pioneering approach to evaluating L-FABP's effectiveness as both a diagnostic and prognostic tool in a pediatric population, the first of its kind to our knowledge. This study contributes valuable data to a field where non-invasive markers are urgently needed for early detection and management of MASLD.

However, the study sample size, though adequate for initial exploration, future longitudinal studies with larger, diverse cohorts are necessary to validate L-FABP's utility across various populations and to monitor disease progression over time. A key limitation of our study is the absence of liver biopsy confirmation due to its invasive nature and ethical concerns in pediatric populations.

## Supporting information

**S1 Fig. ROC Curve AST-ALT Ratio.**
(TIF)

**S2 Fig. ROC Curve Fib3 index.**
(TIF)

**S3 Fig. ROC Curve Fib4 index.**
(TIF)

**S4 Fig. ROC Curve MFS.**
(TIF)

**S5 Table. Dtaset for Submission 2025 JUNE 17.**
(XLSX)

## Acknowledgments

The authors extend their acknowledgments and appreciation to the deanship of postgraduate and scientific research at the following universities: (colleges and deans of Research Affairs); New Giza University, Egypt; National Hepatology and Tropical Medicine Research Institute, Cairo, Egypt; University of Sharjah, UAE; Al Ain University, Abu Dhabi campus, Abu Dhabi-UAE; Dubai Pharmacy College for Girls, Dubai-UAE; Gulf Medical University-UAE; University of Al Maarefa-Dirryia, Riyadh-Saudi Arabia; National University-Sudan, Khartoum, Sudan; Bahri University, Khartoum, Sudan.

## Author contributions

**Conceptualization:** Nadia Al Mazrouei, Asim Ahmed Elnour, Marafi Jammaa Ahmed.

**Data curation:** Mostafa Mohsen ElGharbawy, Mahitab Gamal, Amal Ahmed Mohamed, Semira Abdi Beshir, Ali Awadallah Saeed, Heba Ali ElRamly.

**Formal analysis:** Amal Ahmed Mohamed, Osama Mohamed Ibrahim, Ali Awadallah Saeed.

**Investigation:** Semira Abdi Beshir, Sami Fatehi Abdalla, Heba Ali ElRamly.

**Methodology:** Mostafa Mohsen ElGharbawy, Mahitab Gamal, Shaimaa Emad, Osama Mohamed Ibrahim, Marafi Jammaa Ahmed, Vineetha Menon.

**Project administration:** Mahitab Gamal, Vineetha Menon.

**Resources:** Asim Ahmed Elnour, Semira Abdi Beshir, Sami Fatehi Abdalla, Ali Awadallah Saeed.

**Software:** Shaimaa Emad, Vineetha Menon.

**Validation:** Nadia Al Mazrouei, Shaimaa Emad, Amal Ahmed Mohamed, Asim Ahmed Elnour, Sami Fatehi Abdalla.

**Visualization:** Osama Mohamed Ibrahim, Heba Ali ElRamly.

**Writing – original draft:** Nadia Al Mazrouei, Mostafa Mohsen ElGharbawy.

**Writing – review & editing:** Marafi Jammaa Ahmed.

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
