## [Decision Letter · Decision Letter 0]

14 Feb 2025

PONE-D-24-57390Evaluating Liver Type Fatty Acid Binding Protein as a Diagnostic and Prognostic Biomarker in Non-Alcoholic Fatty Liver Disease in Pediatric PatientsPLOS ONE

Dear Dr. Ahmed,

Thank you for submitting your manuscript to PLOS ONE. After careful consideration, we feel that it has merit but does not fully meet PLOS ONE’s publication criteria as it currently stands. Therefore, we invite you to submit a revised version of the manuscript that addresses the points raised during the review process.

We look forward to receiving your revised manuscript.

Kind regards,

Anna Di Sessa, PhD, MD

Academic Editor

PLOS ONE

Journal Requirements:

Reviewers' comments:

Reviewer's Responses to Questions

**Comments to the Author**

1. Is the manuscript technically sound, and do the data support the conclusions?

Reviewer #1: Yes

Reviewer #2: Partly

2. Has the statistical analysis been performed appropriately and rigorously? 

Reviewer #1: Yes

Reviewer #2: Yes

3. Have the authors made all data underlying the findings in their manuscript fully available?

Reviewer #1: Yes

Reviewer #2: Yes

4. Is the manuscript presented in an intelligible fashion and written in standard English?

Reviewer #1: Yes

Reviewer #2: Yes

5. Review Comments to the Author

Reviewer #1: Comments

1. I would advise providing an epidemiologic data for children such as “with an estimation of 31.8% of all adults” instead of going with “Egypt has a large number of NAFLD pediatrics”.

2. “NAFLD is anticipated to be one of the most common leading causes for liver transplantation” Does this information applies to children too? If liver transplantation is not a huge problem for children, I would advise going with different epidemiologic data

3. I would revise this paragraph- “Liver function tests (LFTs) are needed for diagnosing and following up the liver status. Elevated aspartate transaminase (AST) and alanine transaminase (ALT), typically one to four times the upper limits of normal, are the most frequently observed abnormal laboratory test results. However, Patients with nonalcoholic fatty liver disease, or mild steatosis may have normal or slightly elevated transaminase levels”. Instead, please provide more info if liver enzymes by themselves are good enough to detect NAFLD, or we need more sophisticated laboratory screening parameters. For example, FIB-4 or other currently available biomarkers can be useful in children, or not?

4. I am wondering if the change in terminology of NAFLD to MASLD applies to children as well, or not. If yes, can we update the terminology here?

5. “stages from NAFLD to hepatocellular carcinoma” instead of indicating HCC, maybe it would be better to indicate from NAFLD to cirrhosis, or even with complications such as HCC!

6. I would suggest revising the aim: “The aim of this study was to investigate the potential association between the serum levels of L FABP and the liver functions. Moreover, to assess the use of L-FABP serum concentration as a prognostic and diagnostic biomarker in pediatric NAFLD patients to monitor the extent of fibrosis”. Based on your introduction, you are trying to detect with NAFLD with advanced fibrosis, right? However, “The inclusion criteria were: Asymptomatic patients or with a history of mild or moderate liver disease” you included only mild to moderate patients. So, your biomarker based on the studies are beneficial to detect overall NAFLD, or NAFLD with advanced disease. If advanced disease, you only discussed in your introduction how it is beneficial to catch severe presentations. So, you need to revise the introduction as well.

7. What d you mean you have excluded the patients with “elevated transaminases”? Then, how come you aimed to investigate the relation with elevated trans aminases and your biomarker relation?

8. If those are mild to moderate NAFLD patients, why did you measure AFP values?

Reviewer #2: In this manuscript, Mazrouei et al report the use of L-FABP as a diagnostic and prognostic biomarker for paediatric MASLD. This is a valuable piece of work and builds on findings of others in the field. I would advocate for further clarification of the following points:

(1) What is the rationale for using the term "Non-Alcoholic Fatty Liver Disease", when the international Hepatology community has reached consensus to move to "Metabolic Dysfunction-Associated Steatotic Liver Disease"?

(2) It is not clear how "NAFLD patients" and "controls" were recruited or have been assigned to their respective groups. What were the inclusion criteria for each? Is the "patient" group defined on biopsy, or on non-invasive assessment? If the latter, it does not take anything away from the findings but should be clarified.

How about controls? Have they had any assessment to exclude steatosis, which is, after all, a silent condition?

(3) What criteria were used to define "NAFLD stages"? This should be clarified in the methods section.

(4) The authors put patients into 4 categories based on ultrasonographic assessment. It is not clear what the rationale for these headings is - these headings in turn form the basis for Figure 1. It is not clear why "bright hepatomegaly" and "fatty liver" are listed as different groups. Similarly, does the term "coarse liver" imply that these patients have cirrhosis? This seems unlikely given that the proportion with "coarse liver" is very high, and it does not correlate with the separately reported "NAFLD stages" - although it is not clear how those stages were defined.

(5) Lines 87 - 88 need revising. Younnossi et al, recognised as an authority in the field of MASLD epidemiology, put the global prevalence of MASLD at 30%. The references used refer to global NAFLD and paediatric NAFLD, and it is not clear what exactly is being referred to. I would urge further clarity here.

(6) Line 119 - The aim of the study is to look at the association between L-FABP and "liver functions". It is not clear what the term "liver functions" refers to. If it refers to transaminases, then it should be clarified, although the authors argue later on in the manuscript that transaminases are a poor reflection of liver function per se.

Same comment for the phrase "liver function biomarkers" in line 173.

(7) Line 149 - Could the authors clarify whether these were fasting blood samples or otherwise? It has implications for interpretation of parameters like triglycerides and glucose concentrations.

(8) Line 168 - Minor point - Are the authors sure that Fibroscan is a US brand (rather than French)?

(9) Table 1 - I am not convinced there is a statistical difference in BMI between NAFLD cases and controls. Could the authors clarify the stats?

They should also include standard deviation scores (sds) for BMI, or centiles, as is standard for reporting paediatric parameters.

The row labelled "Sonography" - unclear what was calculated here

(10) Lines 263-267 - Here the authors discuss the utility of ALT in diagnosing and monitoring MASLD, and refer to data from the adult population. Is this similarly true for the paediatric population? I think it is worth being more specific here, as this is the group of interest, and ALT in particular seems to vary across the life course.

(11) Line 301 - Would the authors consider specifiying that L-FABP has "potential utility as a biomarker for the diagnosis of PAEDATRIC NAFLD"?

Finally - I have evaluated the manuscript soleley on scientific merit, rather than language and grammar. If there is the possibility of optimising use of the English language, then it will enhance readability.

6. PLOS authors have the option to publish the peer review history of their article (what does this mean? ). If published, this will include your full peer review and any attached files.

**Do you want your identity to be public for this peer review?** For information about this choice, including consent withdrawal, please see our Privacy Policy .

Reviewer #1: **Yes: ** Akif Altinbas

Reviewer #2: No

---

## [Author Response · Author response to Decision Letter 1]

13 Mar 2025

We sincerely appreciate the time and effort invested by the reviewers in evaluating our manuscript, Evaluating Liver Type Fatty Acid Binding Protein as a Diagnostic and Prognostic Biomarker in Metabolic dysfunction-associated steatotic liver disease in Pediatric Patients ". We are grateful for their insightful comments, which have helped improve the clarity, accuracy, and scientific rigor of our study.

Below, we provide a point-by-point response to each of the reviewers' comments, detailing the modifications made to the manuscript. All changes have been highlighted in the revised document for clarity.

1. I would advise providing an epidemiologic data for children such as “with an estimation of 31.8% of all adults” instead of going with “Egypt has a large number of NAFLD pediatrics”.

1. response

The phrase "Egypt has a large number of NAFLD pediatrics" has been revised to incorporate specific epidemiologic data. Based on available studies, NAFLD prevalence in Egyptian schoolchildren is reported at 15.8%, while among healthy college students, it reaches 31.8%. However, epidemiologic data on pediatric NAFLD in Egypt remains scarce, highlighting the need for further research to investigate this aspect and provide a clearer understanding of disease prevalence and risk factors in this population.

2. “NAFLD is anticipated to be one of the most common leading causes for liver transplantation” Does this information applies to children too? If liver transplantation is not a huge problem for children, I would advise going with different epidemiologic data.

2. response

The original statement regarding NAFLD as a leading cause of liver transplantation was amended based on the recommendation. The revised version now focuses on disease progression in pediatric patients, emphasizing the risk of fibrosis and cirrhosis rather than liver transplantation, which is rare in children. This adjustment ensures the statement is more clinically relevant and aligns with pediatric NAFLD outcomes.

3. I would revise this paragraph- “Liver function tests (LFTs) are needed for diagnosing and following up the liver status. Elevated aspartate transaminase (AST) and alanine transaminase (ALT), typically one to four times the upper limits of normal, are the most frequently observed abnormal laboratory test results. However, Patients with nonalcoholic fatty liver disease, or mild steatosis may have normal or slightly elevated transaminase levels”. Instead, please provide more info if liver enzymes by themselves are good enough to detect NAFLD, or we need more sophisticated laboratory screening parameters. For example, FIB-4 or other currently available biomarkers can be useful in children, or not?

3. response

The paragraph discussing liver function tests (LFTs) for diagnosing and monitoring liver status has been revised to include additional information on the limitations of ALT and AST in detecting NAFLD. The updated version clarifies that while ALT and AST are commonly used markers, they may remain normal in some cases, particularly in mild steatosis. The revision also introduces more sophisticated non-invasive biomarkers, such as FIB-4, NAFLD Fibrosis Score, and Enhanced Liver Fibrosis (ELF) test, discussing their potential utility in pediatric patients. This ensures the text provides a more comprehensive view of diagnostic approaches beyond routine LFTs.

4. I am wondering if the change in terminology of NAFLD to MASLD applies to children as well, or not. If yes, can we update the terminology here?

4. response

Based on the updated guidelines and recent expert perspectives, the terminology change from NAFLD to MASLD applies to pediatric patients as well. The manuscript has been revised accordingly to reflect this updated nomenclature, ensuring consistency with the latest clinical recommendations in pediatric hepatology. The changes also align with the broader classification of Steatotic Liver Disease (SLD), which now includes MASLD and MASH (previously NASH).

5. “stages from NAFLD to hepatocellular carcinoma” instead of indicating HCC, maybe it would be better to indicate from NAFLD to cirrhosis, or even with complications such as HCC!

5. response

The phrase “stages from NAFLD to hepatocellular carcinoma (HCC)” has been revised for better accuracy. Since HCC is rare in pediatric patients, the updated wording now describes the progression from MASLD (formerly NAFLD) to cirrhosis, with potential complications, including HCC. This revision better reflects the natural history of the disease in children, where the primary concern is the development of fibrosis and cirrhosis, which can lead to complications later in life.

6. I would suggest revising the aim: “The aim of this study was to investigate the potential association between the serum levels of L FABP and the liver functions. Moreover, to assess the use of L-FABP serum concentration as a prognostic and diagnostic biomarker in pediatric NAFLD patients to monitor the extent of fibrosis”. Based on your introduction, you are trying to detect NAFLD with advanced fibrosis, right? However, “The inclusion criteria were: Asymptomatic patients or with a history of mild or moderate liver disease” you included only mild to moderate patients. So, your biomarker based on the studies are beneficial to detect overall NAFLD, or NAFLD with advanced disease. If advanced disease, you only discussed in your introduction how it is beneficial to catch severe presentations. So, you need to revise the introduction as well.

6. response

Thank you for your valuable feedback. We acknowledge the need for clarification regarding the study aim and inclusion criteria. By "mild to moderate" in the inclusion section, we intended to encompass all stages of liver fibrosis except for cirrhosis. Consequently, some patients with advanced fibrosis were indeed enrolled. To accurately reflect our study's actual scope and enrollment, we have now amended the description of the inclusion criteria to explicitly state this intent. Additionally, we have updated the exclusion criteria to explicitly mention cirrhosis, ensuring consistency with our actual study population. In light of this, we have revised the introduction to better align with our study's focus—not just on detecting NAFLD in general but also on assessing L-FABP as a biomarker for identifying varying degrees of fibrosis, including advanced fibrosis before cirrhosis develops. This clarification ensures that our study's aim and design are clearly communicated and accurately reflect the population we investigated.

We appreciate your insightful comments, which have helped refine the clarity and precision of our manuscript.

7. What’d you mean you have excluded the patients with “elevated transaminases”? Then, how come you aimed to investigate the relation with elevated trans aminases and your biomarker relation?

7. response

The manuscript has been clarified to specify that only elevated transaminases indicating an acute liver disease were excluded

8. If those are mild to moderate NAFLD patients, why did you measure AFP values?

8. response

AFP was measured to explore whether subtle elevations occur in pediatric NAFLD, even in the absence of malignancy. However, its primary role remains in detecting hepatocellular carcinoma, which is rare in children. (There was a significant difference between control and test groups, P-value <0.0001)

(1) What is the rationale for using the term "Non-Alcoholic Fatty Liver Disease", when the international Hepatology community has reached consensus to move to "Metabolic Dysfunction- Associated Steatotic Liver Disease"?

(1) response

The terminology “Non-Alcoholic Fatty Liver Disease (NAFLD)” was used in the study because, at the time the research was initiated, the transition to “Metabolic Dysfunction-Associated Steatotic Liver Disease (MASLD)” was still under debate within the international hepatology community. Since then, a global consensus has been reached in favor of MASLD as the preferred term. Accordingly, the manuscript has been updated to reflect this terminology change where applicable, ensuring alignment with current hepatology guidelines while acknowledging that the study was conducted during the transitional period in nomenclature.

(2) It is not clear how "NAFLD patients" and "controls" were recruited or have been assigned to their respective groups. What were the inclusion criteria for each? Is the "patient" group defined on biopsy, or on non-invasive assessment? If the latter, it does not take anything away from the findings but should be clarified. How about controls? Have they had any assessment to exclude steatosis, which is, after all, a silent condition?

(2) response

The recruitment process is detailed in the Methods section. MASLD patients were selected based on predefined staging criteria. Inclusion criteria included asymptomatic or mild/moderate liver disease cases, normal INR, age >2.5 to <18 years, and confirmed MASLD diagnosis based on predefined staging. Exclusion criteria included fulminant/acute liver failure, viral hepatitis (HCV, HBV), hepatocellular carcinoma, or refusal of consent. The control group consisted of healthy children with no clinical signs of MASLD. We confirm that controls underwent appropriate assessments, including ultrasound imaging, to exclude hepatic steatosis, given its asymptomatic nature. The extent of fibrosis and the level of steatosis was based on liver brightness in addition to a transient elastography fibroscan (Abdominal examination section) The recruitment process was further clarified and amended in the manuscript (lines 163-167)

(3) What criteria were used to define "NAFLD stages"? This should be clarified in the methods section.

(3) response

We utilized transient elastography (FibroScan®) to determine MASLD stages (F0-F3). This classification is detailed in the Methods section and aligns with established non-invasive fibrosis staging methods. Amended in the manuscript (lines 196-200)

(4) The authors put patients into 4 categories based on ultrasonographic assessment. It is not clear what the rationale for these headings is - hese headings in turn form the basis for Figure 1. It is not clear why "bright hepatomegaly" and "fatty liver" are listed as different groups. Similarly, does the term "coarse liver" imply that these patients have cirrhosis? This seems unlikely given that the proportion with "coarse liver" is very high, and it does not correlate with the separately reported "NAFLD stages" - although it is not clear how those stages were defined.

(4) response

Thank you for your insightful comments. We appreciate the opportunity to clarify the rationale behind the ultrasonographic categories used in our study. These four categories—bright hepatomegaly, coarse liver, fatty liver, and splenomegaly—were defined based on differences in hepatic echotexture, as detailed in the manuscript (Lines 235-237). · Bright hepatomegaly refers to increased echogenicity combined with hepatomegaly. · Fatty liver denotes uniform hepatic steatosis while maintaining a normal liver size. · Coarse liver represents advanced fibrosis but does not indicate cirrhosis. · Splenomegaly is included due to its association with liver disease progression. To avoid any ambiguity, we have clarified these distinctions in the methodology section (Lines 196-200). Importantly, we acknowledge that ultrasonography has limitations in precisely quantifying fibrosis compared to Fibroscan. Therefore, while ultrasonographic patterns were categorized based on echotexture, the degree of fibrosis in our study participants was primarily determined by Fibroscan results. The ultrasonographic findings were considered complementary but not definitive for fibrosis staging. We have ensured that these clarifications are reflected in the revised manuscript to strengthen the coherence between the ultrasonographic classifications and fibrosis staging. We appreciate your feedback, which has helped refine the clarity and precision of our methodology.

(5) Lines 87 - 88 need revising. Younnossi et al, recognised as an authority in the field of MASLD epidemiology, put the global prevalence of MASLD at 30%. The references used refer to global NAFLD and paediatric NAFLD, and it is not clear what exactly is being referred to. I would urge further clarity here.

(5) response

We acknowledge the inconsistency in prevalence estimates. Younossi et al. estimate the global prevalence of MASLD at 30%. We revised the manuscript to clearly differentiate between general MASLD prevalence and pediatric-specific estimates (lines 99-104).

(6) Line 119 - The aim of the study is to look at the association between L-FABP and "liver functions". It is not clear what the term "liver functions" refers to. If it refers to transaminases, then it should be clarified, although the authors argue later in the manuscript that transaminases are a poor reflection of liver function per se. Same comment for the phrase "liver function biomarkers" in line 173.

(6) response

We replaced "liver functions" with a more precise term "hepatic biochemical markers" ensuring clarity regarding which parameters (ALT, AST, albumin, etc.) were evaluated.

(7) Line 149 - Could the authors clarify whether these were fasting blood samples or otherwise? It has implications for interpretation of parameters like triglycerides and glucose concentrations.

(7) response

Yes, blood samples were collected under fasting conditions. We explicitly stated this in the Methods section, as fasting status affects parameters such as triglycerides and glucose levels (now line 177).

(8) Line 168 - Minor point - Are the authors sure that Fibroscan is a US brand (rather than French)?

(8) response

We acknowledge the oversight regarding FibroScan's origin. It is a French brand developed by Echosens. We corrected this in the manuscript.

(9) Table 1 - I am not convinced there is a statistical difference in BMI between NAFLD cases and controls. Could the authors clarify the stats? They should also include standard deviation scores (sds) for BMI, or centiles, as is standard for reporting paediatric parameters. The row labelled "Sonography" - unclear what was calculated here

(9) response

We appreciate the reviewer’s careful examination of Table 1 and for highlighting the statistical difference in BMI between NAFLD cases and controls. We have rechecked our analysis to ensure accuracy and there is a typographical error in the NAFLD Gp BMI. It was just a typo error in the table. The true value for NAFLD gp was (33.69 ± 4.57) not 23.69 This raw was amended to clarify that the statistical testing was performed between L-FABP level and the ultrasonographic categories of patients.

(10) Lines 263-267 - Here the authors discuss the utility of ALT in diagnosing and monitoring MASLD and refer to data from the adult population. Is this similarly true for the paediatric population? I think it is worth being more specific here, as this is the group of interest, and ALT in particular seems to vary across the life course.

(10) response

We appreciate your insightful comments regarding the utility of alanine aminotransferase (ALT) in diagnosing and monitoring Metabolic Dysfunction-Associated Steatotic Liver Disease (MASLD), particularly in the pediatric population. In light of these considerations, we proposed the following revisions to the manuscript to address the variability of ALT levels and its implications for diagnosing and monitoring MASLD in the pediatric population: Revised Text: "ALT levels vary significantly across the lifespan, influenced by factors such as age, sex, and pubertal status. In children and adolescents, ALT levels tend to rise during puberty, with boys exhibiting higher levels than girls. However, relying solely on ALT levels for diagnosing MASLD in the pediatric population has limitations, as a notable proportion of children with biopsy-confirmed MASLD may present with normal ALT levels. Therefore, while ALT can serve as an initial screening tool, particularly in children over 10 years of age with elevated BMI, it should be complemented with other diagnostic modalities. These may include imaging studies, such as ultrasound or magnetic resonance elastography, and the assess

---

## [Decision Letter · Decision Letter 1]

19 Apr 2025

PONE-D-24-57390R1Evaluating Liver Type Fatty Acid Binding Protein as a Diagnostic and Prognostic Biomarker in Metabolic Dysfunction-Associated Steatotic Liver Disease in Pediatric PatientsPLOS ONE

Dear Dr. Ahmed,

Thank you for submitting your manuscript to PLOS ONE. After careful consideration, we feel that it has merit but does not fully meet PLOS ONE’s publication criteria as it currently stands. Therefore, we invite you to submit a revised version of the manuscript that addresses the points raised during the review process.

We look forward to receiving your revised manuscript.

Kind regards,

Anna Di Sessa, PhD, MD

Academic Editor

PLOS ONE

Reviewers' comments:

Reviewer's Responses to Questions

**Comments to the Author**

1. If the authors have adequately addressed your comments raised in a previous round of review and you feel that this manuscript is now acceptable for publication, you may indicate that here to bypass the “Comments to the Author” section, enter your conflict of interest statement in the “Confidential to Editor” section, and submit your "Accept" recommendation.

Reviewer #2: All comments have been addressed

Reviewer #3: All comments have been addressed

Reviewer #4: All comments have been addressed

Reviewer #5: All comments have been addressed

2. Is the manuscript technically sound, and do the data support the conclusions?

Reviewer #2: Yes

Reviewer #3: Yes

Reviewer #4: Yes

Reviewer #5: Partly

3. Has the statistical analysis been performed appropriately and rigorously? 

Reviewer #2: Yes

Reviewer #3: Yes

Reviewer #4: Yes

Reviewer #5: Yes

4. Have the authors made all data underlying the findings in their manuscript fully available?

Reviewer #2: Yes

Reviewer #3: Yes

Reviewer #4: Yes

Reviewer #5: No

5. Is the manuscript presented in an intelligible fashion and written in standard English?

Reviewer #2: Yes

Reviewer #3: Yes

Reviewer #4: Yes

Reviewer #5: Yes

6. Review Comments to the Author

Reviewer #2: Congratulations to the authors for taking suggestions on board and considerably improving their manuscript.

Reviewer #3: The authors have taken the time to answer all the reviewers' questions and in my opinion the manuscript can be accepted for publication.

Reviewer #4: The author has adequately addressed all comments from the previous reviewers. The manuscript is now suitable for publication.

Reviewer #5: Review of "Evaluating Liver Type Fatty Acid Binding Protein as a Diagnostic and Prognostic Biomarker in Metabolic Dysfunction-Associated Steatotic Liver Disease in Pediatric Patients" (PONE-D-24-57390R1).

This study investigated the utility of L-FABP on the markers for MAFLD in pediatric patients. This study showed that the serum L-FABP levels were higher in pediatric patients with MASLD and that the serum L-FABP levels were also higher in patients with fibrosis. This study is potentially interesting; however, several problems need to be solved.

1. The authors state that “liver biopsy is the gold standard in the diagnosis of MASLD”. However, the diagnosis of fatty liver in this study was made using abdominal ultrasound. At least, references should be provided on the utility of abdominal ultrasound and FibroScan� in the diagnosis of MASLD in children.

2. The definitions of “fatty liver” and “MASLD” were not shown.

3. The bright hepatomegaly, coarse liver, fatty liver, and splenomegaly, which are indicated in Figure 1, need to be explained in the method.

4. Table 2. Correlation coefficients, not just p-values, should be shown.

5. To compare the ROC of L-FABP for diagnosis MASLD and that of BMI, fatty liver index, AST/ALT or ALT etc.

6. To compare the utility of L-FABP for diagnosis F2 or F3 stage and that of Fib-4 index, Fib-3 index, MASLD fibrosis score, or AST/ALT, etc.

7. PLOS authors have the option to publish the peer review history of their article (what does this mean? ). If published, this will include your full peer review and any attached files.

**Do you want your identity to be public for this peer review?** For information about this choice, including consent withdrawal, please see our Privacy Policy .

Reviewer #2: No

Reviewer #3: No

Reviewer #4: No

Reviewer #5: **Yes: ** Yoshitaka Hashimoto

---

## [Author Response · Author response to Decision Letter 2]

16 May 2025

We are grateful for the positive reception of our work and the constructive suggestions provided.

We have carefully considered each comment and have made the necessary revisions. Below is a summary of our responses:

1. The authors state that “liver biopsy is the gold standard in the diagnosis of MASLD”. However, the diagnosis of fatty liver in this study was made using abdominal ultrasound. At least, references should be provided on the utility of abdominal ultrasound and FibroScan® in the diagnosis of MASLD in children.

1. Thank you for your insightful observation. We agree that liver biopsy remains the gold standard for diagnosing MASLD; however, due to its invasive nature and limited feasibility in pediatric populations, non-invasive modalities are now commonly used in clinical and research settings.

In our study, abdominal ultrasonography was performed as part of the routine imaging assessment to evaluate general liver morphology and identify overt hepatic steatosis. The diagnosis and staging of MASLD were based on transient elastography (FibroScan® Compact 530),

To address your concern, we have clarified this distinction in the manuscript and added appropriate references supporting the clinical utility of FibroScan® in the evaluation of MASLD in pediatric populations (References 23–25).

2. The definitions of “fatty liver” and “MASLD” were not shown.

2. Thank you for pointing that out. The definitions of “fatty liver” and “MASLD” were previously missing and have now been added at line 90 for clarity and completeness.

3. The bright hepatomegaly, coarse liver, fatty liver, and splenomegaly, which are indicated in Figure 1, need to be explained in the method.

3. We appreciate your suggestion and have revised the Methods section accordingly to include clear definitions of the ultrasound findings mentioned in Figure 1. Specifically, we now explain the terms “bright hepatomegaly,” “coarse liver,” “fatty liver,” and “splenomegaly” as part of the routine sonographic evaluation. (line 202)

4. Table 2. Correlation coefficients, not just p-values, should be shown.

4. Pearson’s partial correlation coefficient is now added to the table (for univariable analysis while adjusting for group effect).

5. To compare the ROC of L-FABP for diagnosis MASLD and that of BMI, fatty liver index, AST/ALT or ALT etc.

5. Thank you for this important comment. We acknowledge the potential interest in comparing L-FABP to other non-invasive markers such as BMI, fatty liver index (FLI), AST/ALT ratio, and ALT alone. However, as discussed in the subsequent comment, the utility and diagnostic validity of these parameters in pediatric populations remain limited and not well-established.

For example, the Fatty Liver Index was developed and validated in adults and has not been appropriately adapted or validated for use in children. Similarly, indices like AST/ALT ratio and isolated ALT measurements lack sufficient sensitivity and specificity for diagnosing MASLD or staging fibrosis in the pediatric setting. Given these limitations, we chose not to include these comparisons in our ROC analysis to avoid drawing potentially misleading conclusions.

6. To compare the utility of L-FABP for diagnosis F2 or F3 stage and that of Fib-4 index, Fib-3 index, MASLD fibrosis score, or AST/ALT, etc.

6. Thank you for your insightful comment regarding the use of non-invasive fibrosis scores such as FIB-4, APRI, and NFS in the evaluation of MASLD. While these scoring systems are widely used and validated in adult populations, their utility in children remains limited and lacks sufficient validation.

The FIB-4 index—although well-established in adults—has reduced reliability in certain populations, particularly in individuals under 35 years of age, where it tends to underperform. This is especially relevant in pediatric cohorts, where age-related physiological differences affect the accuracy of these indices. Additionally, FIB-4’s diagnostic value is limited in the intermediate range and may result in false positives in populations with a lower prevalence of advanced fibrosis, further limiting its clinical applicability in younger age groups.

Given these constraints, and in alignment with current literature, we opted not to rely on FIB-4 or similar adult-derived scores in our pediatric MASLD study. https://www.journal-of-hepatology.eu/article/S0168-8278(24)00329-5/fulltext

https://journals.lww.com/ajg/abstract/2017/05000/age_as_a_confounding_factor_for_the_accurate.21.aspx

We appreciate your thoughtful feedback and trust this addresses your concern.

---

## [Decision Letter · Decision Letter 2]

30 May 2025

PONE-D-24-57390R2Evaluating Liver Type Fatty Acid Binding Protein as a Diagnostic and Prognostic Biomarker in Metabolic Dysfunction-Associated Steatotic Liver Disease in Pediatric PatientsPLOS ONE

Dear Dr. Ahmed,

Thank you for submitting your manuscript to PLOS ONE. After careful consideration, we feel that it has merit but does not fully meet PLOS ONE’s publication criteria as it currently stands. Therefore, we invite you to submit a revised version of the manuscript that addresses the points raised during the review process.

We look forward to receiving your revised manuscript.

Kind regards,

Anna Di Sessa, PhD, MD

Academic Editor

PLOS ONE

Reviewers' comments:

Reviewer's Responses to Questions

**Comments to the Author**

1. If the authors have adequately addressed your comments raised in a previous round of review and you feel that this manuscript is now acceptable for publication, you may indicate that here to bypass the “Comments to the Author” section, enter your conflict of interest statement in the “Confidential to Editor” section, and submit your "Accept" recommendation.

Reviewer #5: All comments have been addressed

2. Is the manuscript technically sound, and do the data support the conclusions?

Reviewer #5: Partly

3. Has the statistical analysis been performed appropriately and rigorously? 

Reviewer #5: Yes

4. Have the authors made all data underlying the findings in their manuscript fully available?

Reviewer #5: Yes

5. Is the manuscript presented in an intelligible fashion and written in standard English?

Reviewer #5: Yes

6. Review Comments to the Author

Reviewer #5: This reviewer understands that BMI, fatty liver index, AST/ALT or ALT, Fib-4 index, Fib-3 index, and MASLD fibrosis score are useful in adults, but their usefulness in children has not been clearly established. The purpose of this study is to show the utility of L-FABP; thus, the authors should be compared these markers with L-FABP to clarify this point.

7. PLOS authors have the option to publish the peer review history of their article (what does this mean? ). If published, this will include your full peer review and any attached files.

**Do you want your identity to be public for this peer review?** For information about this choice, including consent withdrawal, please see our Privacy Policy .

Reviewer #5: **Yes: ** Yoshitaka Hashimoto

---

## [Author Response · Author response to Decision Letter 3]

17 Jun 2025

Response to Reviewers

Point to point response

Dear Respected Editor

Good day

Thank you for the constructive comments, hereby were our responses.

Comment

Reviewer's Responses to Questions from Comments to the Author

1. If the authors have adequately addressed your comments raised in a previous round of review and you feel that this manuscript is now acceptable for publication, you may indicate that here to bypass the “Comments to the Author” section, enter your conflict of interest statement in the “Confidential to Editor” section, and submit your "Accept" recommendation.

Reviewer #5: All comments have been addressed.

Response

Thanks

2. Is the manuscript technically sound, and do the data support the conclusions?

Reviewer #5: Partly

Response

We assume this assessment is primarily based on the concern raised in point 6. In response, we have addressed this point thoroughly in our reply to comment 6. We hope that this added analysis satisfactorily addresses the reviewer’s concern and affirms the methodological rigor and data-supported conclusions of the study.

3. Has the statistical analysis been performed appropriately and rigorously?

Reviewer #5: Yes

Response

Thanks

4. Have the authors made all data underlying the findings in their manuscript fully available?

Reviewer #5: Yes

Response

Thanks

5. Is the manuscript presented in an intelligible fashion and written in standard English?

Reviewer #5: Yes

Response

Thanks

6. Review Comments to the Author

Reviewer #5: This reviewer understands that BMI, fatty liver index, AST/ALT or ALT, Fib-4 index, Fib-3 index, and MASLD fibrosis score are useful in adults, but their usefulness in children has not been clearly established. The purpose of this study is to show the utility of L-FABP; thus, the authors should be compared these markers with L-FABP to clarify this point.

Comment

Response

We thank the reviewer for highlighting the importance of ensuring scientific rigor and drawing conclusions supported by robust data. In response, we would like to clarify that non-invasive fibrosis scores, including AST/ALT, Fib-4 index, Fib-3 index, and MAFLD fibrosis score, were indeed calculated and analyzed in our study. These scores were used to provide a comparative context for evaluating the diagnostic performance of L-FABP in detecting liver fibrosis among the pediatric MASLD population. Despite the fact that these indices were originally developed and validated in adult populations, we performed ROC curve analyses comparing them to L-FABP. This was done to assess whether L-FABP offers any additional diagnostic utility or improved performance in a pediatric setting. The results, which are now included in the revised manuscript [lines 296-302, 403-416, 422-424], show that L-FABP demonstrates superior diagnostic accuracy compared to the aforementioned scores in identifying children with stage of fibrosis. We believe that including this comparative analysis strengthens the validity of our conclusions and demonstrates that our findings are drawn from technically sound and appropriately controlled data.

7. PLOS authors have the option to publish the peer review history of their article (what does this mean?). If published, this will include your full peer review and any attached files.

YES.

---

## [Decision Letter · Decision Letter 3]

22 Jul 2025

PONE-D-24-57390R3Evaluating Liver Type Fatty Acid Binding Protein as a Diagnostic and Prognostic Biomarker in Metabolic Dysfunction-Associated Steatotic Liver Disease in Pediatric PatientsPLOS ONE

Dear Dr. Ahmed,

Thank you for submitting your manuscript to PLOS ONE. After careful consideration, we feel that it has merit but does not fully meet PLOS ONE’s publication criteria as it currently stands. Therefore, we invite you to submit a revised version of the manuscript that addresses the points raised during the review process.

We look forward to receiving your revised manuscript.

Kind regards,

Anna Di Sessa, PhD, MD

Academic Editor

PLOS ONE

Journal Requirements:

Additional Editor Comments:

I believe that the paper would benefit from a major revision before considering it for publication.

Reviewers' comments:

Reviewer's Responses to Questions

**Comments to the Author**

1. If the authors have adequately addressed your comments raised in a previous round of review and you feel that this manuscript is now acceptable for publication, you may indicate that here to bypass the “Comments to the Author” section, enter your conflict of interest statement in the “Confidential to Editor” section, and submit your "Accept" recommendation.

Reviewer #4: All comments have been addressed

Reviewer #6: (No Response)

2. Is the manuscript technically sound, and do the data support the conclusions?

Reviewer #4: Yes

Reviewer #6: Partly

3. Has the statistical analysis been performed appropriately and rigorously? 

Reviewer #4: Yes

Reviewer #6: No

4. Have the authors made all data underlying the findings in their manuscript fully available?

Reviewer #4: Yes

Reviewer #6: No

5. Is the manuscript presented in an intelligible fashion and written in standard English?

Reviewer #4: Yes

Reviewer #6: Yes

6. Review Comments to the Author

Reviewer #4: I think the manuscript suitable for publication and have no concerns regarding research integrity, ethics, or potential dual publication.

Reviewer #6: The aim of this study is intriguing. However, this Reviewer has several major issues that should be addressed.

MAJOR REVISIONS

- What about on-going pharmacological treatment? In discussion you mentioned UDCA as possible modifier of ALT, but no information are given about treatment in study population.

- What about genetic metabolic diseases? Were they investigated/excluded?

- Aim of the study: please revise the sentences at lines 139-144. They sound redundant, although the first sentence does not explain the kind of population “… in a the pediatric population”

- MASLD STAGES: how were they assessed? It seems that they refer to fibrosis stages. If so, please change the definition of MASLD stages into fibrosis stages (and consequently, modify X axis legend in figure 2), otherwise provide ref for such classification.

- SONOGRAPHIC FINDINGS: why did you use such classification? Are you sure it reflects the entity of liver steatosis and fibrosis? Please provide reference for such risk stratification.

- TABLE 2: how correlations with MASLD stages and Ultrasonographic Category were performed? Pearson is for continuous variables.

- A unique figure in the manuscript with different ROC curves for other NITS could be useful instead of several supplementary figures could be useful.

- liver biospy were not performed, please mention it in the Limitation study section

MINOR REVISIONS

- REF 26: explications for Youden Index should be provided in methods section, not results

- Please provide ref for Chen, Chen, et al. (line 319)

- Please provide reference for sentence at lines 129-130

- Sintax mistakes: lines 123-125; line 138; line 289 “being .. diagnosed MASLD”; along several lines there are unexplicable brackets (e.g. (MASLD) in the abstract, (F0-F1) at line 306)

7. PLOS authors have the option to publish the peer review history of their article (what does this mean? ). If published, this will include your full peer review and any attached files.

**Do you want your identity to be public for this peer review?** For information about this choice, including consent withdrawal, please see our Privacy Policy .

Reviewer #4: No

Reviewer #6: **Yes: ** Lucilla Crudele

---

## [Author Response · Author response to Decision Letter 4]

28 Jul 2025

Response to Reviewers

Point to point response

Dear Respected Editor

Good day

Thank you for the constructive comments, hereby were our responses.

Comment

Reviewer's Responses to Questions from Comments to the Author

1. If the authors have adequately addressed your comments raised in a previous round of review and you feel that this manuscript is now acceptable for publication, you may indicate that here to bypass the “Comments to the Author” section, enter your conflict of interest statement in the “Confidential to Editor” section, and submit your "Accept" recommendation.

Reviewer #6: (No Response)

2. Is the manuscript technically sound, and do the data support the conclusions?

Reviewer #5: Partly

Response

We assume this assessment is primarily based on the concern raised in points 3 and 4. In response, we have addressed this point thoroughly in our reply to comments 3 and 4. We hope that this added analysis satisfactorily addresses the reviewer’s concern and affirms the methodological rigor and data-supported conclusions of the study.

3. Has the statistical analysis been performed appropriately and rigorously?

Reviewer #6: No

Response

We assume this assessment is primarily based on the concern raised in point 6. In response, we have addressed this point thoroughly in reply to comment 6. We hope that this added analysis satisfactorily addresses the reply and affirms the statistical rigor and appropriateness.

4. Have the authors made all data underlying the findings in their manuscript fully available?

The PLOS Data policy requires authors to make all data underlying the findings described in their manuscript fully available without restriction, with rare exception (please refer to the Data Availability Statement in the manuscript PDF file). The data should be provided as part of the manuscript or its supporting information or deposited to a public repository. For example, in addition to summary statistics, the data points behind means, medians and variance measures should be available. If there are restrictions on publicly sharing data—e.g. participant privacy or use of data from a third party—those must be specified.

Reviewer #6: No

Response

We confirm that all raw data underlying the findings of this study have been provided as a supplementary file accompanying the manuscript. Additionally, a Data Availability Statement has been added to the manuscript, clarifying that all relevant data are fully available without restriction and can be accessed in the supplementary material.

5. Is the manuscript presented in an intelligible fashion and written in standard English?

Reviewer #6: Yes

Response

Thanks

6. Review Comments to the Author

Reviewer #6: The aim of this study is intriguing. However, this Reviewer has several major issues that should be addressed.

Response to MAJOR REVISIONS

1. What about on-going pharmacological treatment? In discussion you mentioned UDCA as possible modifier of ALT, but no information are given about treatment in study population.

Response

We appreciate the reviewer’s valuable comment regarding pharmacological treatment. We have now clarified in the Patients and Study Design section that all enrolled patients received treatment protocols as prescribed by their managing physicians, with no interference from the research team as this was an observational non-interventional study. Importantly, none of the participants were receiving ursodeoxycholic acid (UDCA) or other hepatoprotective agents that could influence ALT or L-FABP levels.

Additionally, we would like to emphasize that by the time of the study, there were currently no pharmacological agents approved specifically for the treatment of MASLD (previously NAFLD) in the pediatric population. The cornerstone of management remains lifestyle modification, which includes gradual weight reduction (if needed), adherence to a balanced diet low in added sugars and saturated fats, and engagement in regular physical activity. These interventions were encouraged as part of routine clinical care for the patients but were not controlled or modified by the study team. (Lines 164-169)

2. What about genetic metabolic diseases? Were they investigated/excluded?

Response

Thank you for this observation. We confirm that children with suspected or confirmed genetic or metabolic liver diseases were excluded from the study through medical history review and prior diagnostic testing. (Lines 174-176).

3. Aim of the study: please revise the sentences at lines 139-144. They sound redundant, although the first sentence does not explain the kind of population “… in a the pediatric population”

Response

We agree and have revised the aim section for clarity and precision. (lines 137-139)

4. MASLD STAGES: how were they assessed? It seems that they refer to fibrosis stages. If so, please change the definition of MASLD stages into fibrosis stages (and consequently, modify X axis legend in figure 2), otherwise provide ref for such classification.

Response

Yes, the “MASLD stages” referred to in the manuscript correspond to fibrosis stages as determined by transient elastography (FibroScan®). The terminology was corrected throughout the manuscript and the X-axis of Figure 2 was updated.

5. SONOGRAPHIC FINDINGS: why did you use such classification? Are you sure it reflects the entity of liver steatosis and fibrosis? Please provide reference for such risk stratification.

Response

We thank the reviewer for raising this point. Ultrasound classification was included in our study as one of the basic, routine, non-invasive tools for initial hepatic assessment. It is widely used for screening liver steatosis and fibrosis due to its accessibility, safety, low cost, and non-invasiveness. In pediatric MASLD, ultrasound serves as a first-line imaging modality to identify features such as hepatic echogenicity and hepatomegaly.

However, we would like to emphasize that ultrasound findings were not the sole or determining measure for staging liver fibrosis in this study. Fibrosis stages were assessed using transient elastography (FibroScan®), which offers a more objective and quantifiable evaluation of liver stiffness. Ultrasound was used primarily to describe hepatic morphology and support the overall clinical assessment.

This was clarified in lines 246 and 247. Also, references were added

6. TABLE 2: how correlations with MASLD stages and Ultrasonographic Category were performed? Pearson is for continuous variables.

Response

We thank the reviewer. Pearson’s correlation was applied only to numeric variables. Consequently, MASLD stages and ultrasonographic categories—being categorical in nature—were not included in the Pearson correlation analysis, which is why “NA” appears in Table 2 for these variables. Instead, their associations with serum L FABP were assessed using analysis of variance (ANOVA) on top of regression modeling.

7. A unique figure in the manuscript with different ROC curves for other NITS could be useful instead of several supplementary figures could be useful.

Response

We thank the reviewer for this helpful suggestion. We have addressed it by consolidating the ROC curves of the non invasive tests (NITs) into one figure.

8. liver biopsy were not performed, please mention it in the Limitation study section

Response

Thanks for the comment. It was added as a limitation. (Line 456-457)

MINOR REVISIONS

• REF 26: explications for Youden Index should be provided in methods section, not results

Response

The reference was added in the method section (line 224)

• Please provide ref for Chen, Chen, et al. (line 319)

Response

It is Reference 34 (Chen Z-w, Chen L-y, Dai H-l, Chen J-h, Fang L-z. Relationship between alanine aminotransferase levels and metabolic syndrome in nonalcoholic fatty liver disease. J Zhejiang Univ Sci B. 2008;9:616-22.) in line 325

• Please provide reference for sentences at lines 129-130

Response

Added after that sentence instead of at the end of the paragraph (Reference 20)

• Sintax mistakes: lines 123-125; line 138; line 289 “being .. diagnosed MASLD”; along several lines there are unexplicable brackets (e.g. (MASLD) in the abstract, (F0-F1) at line 306)

Response

We thank the reviewer for their careful attention to the language and formatting. The identified issues have been thoroughly reviewed and corrected as follows:

• Lines 123–125:

The sentence describing L-FABP has been revised for grammatical accuracy and clarity. The corrected version now reads:

“Liver-type fatty acid-binding protein (L-FABP) is an antioxidant protein endogenously synthesized in the liver and the proximal tubular epithelial cells of the kidney, but not in skeletal or cardiac muscle.”

• Line 138:

A missing article was corrected. The phrase “in paediatric population” was revised to:

“in the pediatric population”

• Line 289:

The phrasing “being MASLD” was revised to the grammatically correct form:

“being diagnosed with MASLD”

• Unexplained Brackets:

All instances of unnecessary or unexplained parentheses—such as around “(MASLD)” in the abstract—have been reviewed and removed unless part of a formal abbreviation introduction. Technical bracketed terms like (F0–F1) were retained only when used appropriately to denote fibrosis staging.

These amendments have been reflected in the re

---

## [Editor Report · Decision Letter 4]

16 Sep 2025

Evaluating Liver Type Fatty Acid Binding Protein as a Diagnostic and Prognostic Biomarker in Metabolic Dysfunction-Associated Steatotic Liver Disease in Pediatric Patients

PONE-D-24-57390R4

Dear Dr. Ahmed,

We’re pleased to inform you that your manuscript has been judged scientifically suitable for publication and will be formally accepted for publication once it meets all outstanding technical requirements.

Kind regards,

Anna Di Sessa, PhD, MD

Academic Editor

PLOS ONE
---

## [Editor Report · Acceptance letter]

PONE-D-24-57390R4

PLOS ONE

Dear Dr. Ahmed,

I'm pleased to inform you that your manuscript has been deemed suitable for publication in PLOS ONE. Congratulations! Your manuscript is now being handed over to our production team.

Kind regards,

on behalf of

Dr. Anna Di Sessa

Academic Editor

PLOS ONE